# Preserving Label Correlation for Multi-label Text Classification by Prototypical Regularizations

## Abstract

Multi-label text classification (MLTC) aims to assign multiple relevant labels to a given sentence. An inherent challenge of MLTC is capturing label correlations compared with multi-class text classification. Existing MLTC models primarily focus on leveraging correlation information but often overlook the common issue of overfitting. Meanwhile, plug-and-play regularization methods struggle to preserve correlations effectively. In this paper, we distinguish two types of label correlations: explicit co-occurring correlation and implicit semantic correlations, and propose two regularization methods based on prototypical label embeddings for two correlation preservation, respectively. Specifically, we first generate the prototypical label embedding of multiple co-occurred labels as an intermediate. We then apply a prototypical label regularization on the distance between the sentence embedding and corresponding prototypical label embedding to alleviate the over-alignment issue caused by binary cross entropy loss and facilitate explicit correlation preservation. We finally extend the vanilla Mixup, which solely mixes multi-hot labels, on prototypical label embedding mixing to promote implicit correlation preservation. Empirical studies show the effectiveness of our regularization methods.

## CCS Concepts

• **Computing methodologies** → **Regularization**.

## Keywords

Multi-label Text Classification, Prototypical Label, Mixup

**ACM Reference Format:**
Anonymous Author(s). 2018. Preserving Label Correlation for Multi-label Text Classification by Prototypical Regularizations. In *Proceedings of Make sure to enter the correct conference title from your rights confirmation emai (Conference acronym 'XX)*. ACM, New York, NY, USA, 11 pages. https://doi.org/XXXXXXX.XXXXXXX

## 1 Introduction

Multi-label text classification (MLTC) aims to assign multiple labels to a text instance, attracting increasing research interests in recent years [12, 14, 21, 31, 33, 35, 37]. Compared with the standard multi-class text classification, MLTC is inherently more challenging due to the correlations of its intrinsic labels [23, 35]. Numerous

graph-based [21, 36] and attention-based [29, 33] methods are introduced to inject this correlation into dense label embeddings. When training data is abundant and each label is well-represented by numerous instances, these models can effectively learn label correlations on their own. However, in real-world scenarios, acquiring thorough multi-label annotations is significantly more difficult compared to single-label annotations [5]. This practical difficulty of MLTC prevents the model from fully capturing the semantic correlations among labels, leading overfitting problem. Common plug-and-play regularization methods, such as weight decay [15], Dropout [26], and layer normalization [2], typically address overfitting but tend to overlook the crucial label correlation information in MLTC. To overcome this limitation, we propose an innovative regularization framework specifically designed to preserve label correlations while more effectively mitigating overfitting in MLTC.

Notably, label correlations in MLTC can be divided into two categories: explicit intra-instance correlations and implicit inter-instance semantic correlations. Explicit correlation refers to relationships between labels associated with the same sentence (label co-occurrence), whereas implicit correlation involves labels that are not linked to the same sentence but share semantic associations. For example, as shown in Figure 1, 'Machine Learning' and 'Image Classification' are associated with the same sentence, reflecting explicit correlation. On the other hand, 'Image Classification' and 'Text Classification' do not appear in the same sentence but are semantically related, indicating implicit correlation.

In this paper, we aim to simultaneously preserve both types of correlations while addressing the overfitting problem. We utilize prototypical embeddings of co-occurring labels as a bridge and apply two regularization methods based on this embedding to maintain explicit and implicit correlations separately. Note that label relationships in MLTC can be preserved through distributed label embeddings. Previous MLTC models [10, 32] improve label embeddings using prototypes, which are typically computed by averaging related text embeddings, with each prototypical embedding vector representing a single label. In contrast, we argue that correlations among labels are maintained within the prototypical embedding of co-occurring labels—the subset of label embeddings for a given sentence. Compressing co-occurring labels into one prototypical label embedding not only preserves explicit correlations but also enables the application of Mixup [8, 34], which can effectively exploit implicit correlations.

More precisely, existing training objectives fail to preserve explicit correlation because they use multiple binary cross-entropy (BCE) losses, which may push incorrect label embeddings away from their cluster while forcing correct label embeddings to align with their corresponding sentence representations, leading to an over-alignment problem. To mitigate this, we propose applying $L_2$ regularization on the distance—also known as mean squared error (MSE)—between the prototypical label embedding and its sentence

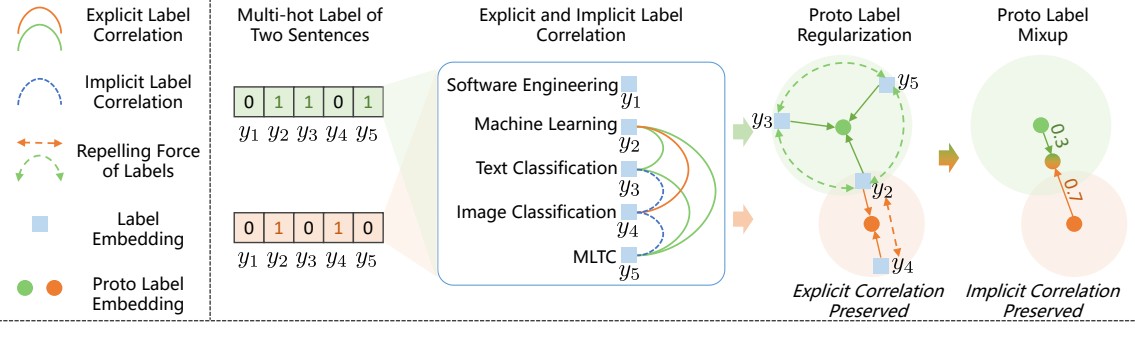

**Figure 1: The illustration of ProtoMix, featuring two sentences with labels $y_2$, $y_3$, $y_5$, and labels $y_2$, $y_4$, respectively. In this case, $y_2$, $y_3$ and $y_5$ are related to the same sentence, enjoying explicit correlation. Similarly, $y_2$ and $y_4$ are as well. Considering the label semantic information, such as 'Image Classification' and 'Text Classification', although the pairs $y_4$ and $y_3$, as well as $y_4$ and $y_5$, are not directly linked to the same sentence, they still possess an implicit correlation. Noted, 'Proto' represents 'Prototypical'.**

embedding. Our theoretical analysis (detailed in Section 4.2) demonstrates that our new loss can be decomposed into two terms: one that introduces a repelling force among the ground-truth labels within the same cluster, and another that is similar to the original BCE loss. This design helps maintain separation among label embeddings, preventing collapse or over-alignment. By balancing the attraction and repelling forces with the BCE loss, the model can enrich the diversity of labels and preserve the correct explicit correlation among related labels.

For implicit correlation preservation, we apply Mixup-based regularization techniques. Mixup generates in-between samples by linearly interpolating both input representations and labels, showing promising results in capturing implicit correlations [9, 27]. Applying vanilla Mixup to MLTC, which interpolates multi-hot label embeddings, may reduce overfitting but often fails to capture the interdependence among labels. To tackle this, we extend the vanilla Mixup framework from label mixing to label representation mixing, i.e. prototypical label mixing, for MLTC tasks. In particular, given a mixing ratio $\lambda$ drawn from a Beta distribution, we apply Mixup to the sentence embeddings and their corresponding prototypical label embeddings by linear interpolation. This approach generates in-between samples that enhance the preservation of implicit semantic label correlations.

These two regularization methods, namely $L_2$ regularization and Mixup, are applied to prototypical label embeddings for correlation preservation, forming a flexible regularization framework named ProtoMix. Figure 1 illustrates this framework. The contributions of our ProtoMix are outlined below:

- We analyze the over-alignment problem induced by BCE loss and justify the use of MSE loss on prototypical label embeddings, which promotes increased diversity among co-occurred labels and leverages explicit label correlation.
- To the best of our knowledge, ProtoMix is the first attempt at exploiting Mixup in MLTC, which applies Mixup on both multi-hot label embedding and prototypical label embedding to preserve the implicit label correlation.
- Empirical studies on the three benchmarking MLTC datasets using various encoders, as well as other analyses confirm the effectiveness of our proposed ProtoMix model.

## 2 Related Work

There exist many MLTC models, and most of them mainly focus on learning enhanced label representation to formulate label correlations to improve classification performance. We divide these methods into three categories: attention-based models, graph-based models, and other representative models.

*Attention-based Methods for MLTC.* This kind of model utilizes the attention mechanism to capture label-specific document representation, which formulates the relationship between texts and labels explicitly. [4] combines word-vector self-attention with AutoEncoder to capture label and feature dependencies. [6] introduces attention to qualitative matching signals between textual contents and the classes to enhance label representation. [29] incorporates the self-attention mechanism to identify label-specific text representations based on the content information. [35] learns correlation-guided representation by jointly learning words and labels in the same space, so as to capture high-order label-label and context-label correlations for MLTC.

*Graph-based Methods for MLTC.* Capturing label correlation is a key challenge and an essential factor for MLTC. Since label dependencies can be represented as a graph, some graph-based methods have been proposed to address this issue. [36] utilizes deep random walk to establish the label representation via the label co-occurrence graph. [21] constructs a dual graph convolution network to capture adaptive interactions of label representation. [33] generates a shallow and wide probabilistic label tree (PLT) to handle millions of labels, especially for tail labels.

*Other Representative Methods for MLTC.* In addition to these two kinds of methods, many novel models effectively solve MLTC tasks. [31] applies a sequence generation model with a novel decoder structure for MLTC. [28] employs contrastive learning enhanced nearest neighbor mechanism for MLTC problem. [32] utilizes a prototypical network to model input instances for label embedding generation. [30] proposes a Head-to-Tail Network to transfer the meta-knowledge from the data-rich head labels to data-poor tail labels. [37] proposes a variational continuous label distribution learning framework, enabling the extraction of hidden information within the observable logical labels.

## 3   Preliminaries

Consider a training set $D = \{(\mathbf{X}_i, \mathbf{Y}_i)\}_{i=1}^{I}$ with $I$ instances. Let $\mathbf{X}_i = [w_1, w_2, ..., w_N]$ be a sentence of $N$ words, and let $\mathbf{Y}_i = \{0, 1\}^K \in \mathbb{R}^K$ be a multi-hot vector, where $K$ is the total number of labels. In a typical multi-label text classification (MLTC) model, the sentence $\mathbf{X}_i$ is encoded into a sentence embedding $\mathbf{S}_i \in \mathbb{R}^H$. The encoding process can be performed using various sentence encoding models $E$, defined as:

$$\mathbf{S}_i = E(\mathbf{X}_i). \qquad (1)$$

The model then uses a fully-connected layer with parameter $\mathbf{W} \in \mathbb{R}^{H \times K}$ to determine the label of sentence as $\hat{\mathbf{Y}}_i$, which is formulated as follows:

$$\hat{\mathbf{Y}}_i = \sigma(\mathbf{S}_i^\top \mathbf{W}), \qquad (2)$$

where $\sigma$ is the sigmoid activation function, and $\hat{\mathbf{Y}}_i$ is the output of a fully connected (FC) classification layer qualifying the matching and interaction between sentence and label representations. In mathematical terms, the FC layer's parameter matrix $\mathbf{W}$ can be interpreted as label representations, where each column is associated with a specific label, also known as label embeddings [6]. As a result, the FC classifier's operation is also interpreted via the dot product mechanism.

Conventional MLTC models treat this task as a multiple binary classification problem and compute the BCE loss for the prediction $\hat{\mathbf{Y}}_i$ in relation to $\mathbf{Y}_i$. The overall loss is defined by the summation of individual BCE losses for each one-hot label, as shown below:

$$\mathcal{L}_C(\mathbf{Y}_i, \hat{\mathbf{Y}}_i) = -\sum_{k=1}^{K} \mathbf{Y}_i^{[k]} \log(\hat{\mathbf{Y}}_i^{[k]}) + (1 - \mathbf{Y}_i^{[k]}) \log(1 - \hat{\mathbf{Y}}_i^{[k]}), \quad (3)$$

where $[\cdot]$ denotes the element selection function, for example, $\mathbf{Y}_i^{[k]}$ and $\hat{\mathbf{Y}}_i^{[k]}$ denote the $k^{\text{th}}$ element (or one-hot vector without ambiguity) of vector $\mathbf{Y}_i$ and $\hat{\mathbf{Y}}_i$, respectively.

## 4   ProtoMix

In the MLTC task, capturing label correlation is essential for effective label representation, particularly when compared to multi-class text classification [36]. This correlation should not be neglected in the development of regularization techniques. In our framework, we distinguish between two types of label relationships: explicit co-occurring correlations and implicit semantic correlations. To preserve both types of correlation information, we introduce two regularization methods that utilize intermediate prototypical label embeddings.

Firstly, we generate prototypical label embeddings for sets of co-occurring labels. Concretely, we propose a sentence-attentive mechanism to construct these prototypical label embeddings, taking into account the intrinsic semantic relationships between sentences and labels. We then apply a $L_2$ regularization on the distance between the prototypical label embeddings and sentence embeddings to mitigate the over-alignment issue caused by BCE loss, therefore preserving explicit label correlations. Additionally, we extend the vanilla Mixup technique by transitioning from multi-hot label mixing to prototypical label embedding mixing. This method leverages implicit label correlations by generating interpolated samples.

## 4.1   Prototypical Label Gerneration

In few-shot text learning scenarios, Prototypical Network [25] represents class prototype embeddings as the geometric centroid of the corresponding sentence cluster in the embedding space. This centroid serves as a better representation as it considers all related sentences of a class. In the context of MLTC, a sentence is associated with a collection of labels, and the centroid derived from these co-occurring labels can significantly enhance label representation by effectively capturing label correlation. Therefore, prototypical label embeddings provide an elegant solution for label representation in the MLTC task. In the remaining of this subsection, we first introduce a basic average centroid approach for prototypical label embeddings, and then design sentence-attentive prototypical label embeddings to better capture the correlations between the sentences and their corresponding multiple labels.

*4.1.1   Average Prototypical Label Embedding.* Averaging is a simple yet effective way to obtain prototype representations. Hence, we average the subset of labels associated with a sentence to preserve label co-occurrence. Formally, we define a one-hot label embedding function $f_\omega : \mathbb{R}^K \to \mathbb{R}^H$ with learnable parameters $\omega$, mapping each label into a joint space with the corresponding text. Given a sentence $\mathbf{X}_i$ and its multi-hot label vector $\mathbf{Y}_i \in \mathbb{R}^K$, the corresponding prototypical label embedding $\mathbf{P}_i \in \mathbb{R}^H$ is calculated as:

$$\mathbf{P}_i = \frac{1}{|\mathcal{K}_i|} \sum_{k \in \mathcal{K}_i} f_\omega(\mathbf{Y}_i^{[k]}), \qquad (4)$$

where $\mathcal{K}_i$ is the set of indices for the ground truth one-hot labels associated with $\mathbf{X}_i$, and $|\mathcal{K}_i|$ denotes the cardinality of this set.

Recall the FC layer defined in Eq.2, $\hat{\mathbf{Y}}_i = \sigma(\mathbf{S}_i^\top \mathbf{W})$. We view $\mathbf{W} \in \mathbb{R}^{H \times K}$ as the label embedding matrix, where each column vector $\mathbf{W}_{\cdot k} \in \mathbb{R}^H$ is the embedding of the $k^{\text{th}}$ label. The logits inside the sigmoid function result from the dot product operation $\mathbf{S}_i^\top \mathbf{W} \in \mathbb{R}^K$ between the sentence embedding and all $K$ label embeddings. For a specific sentence $\mathbf{X}_i$ in multi-class classification, the label embedding can be retrieved by taking the dot product of its one-hot label $\mathbf{Y}_i^{[k]}$ with the weight matrix $\mathbf{W}$, i.e., $\mathbf{W}\mathbf{Y}_i^{[k]}$. Similarly, for the multi-label case, we define the prototypical label embedding $\mathbf{P}_i$ of $\mathbf{X}_i$ and its multi-hot label $\mathbf{Y}_i$ as:

$$\mathbf{P}_i = \frac{1}{\sum_{k \in \mathcal{K}_i} \mathbf{Y}_i^{[k]}} \mathbf{W}\mathbf{Y}_i = \frac{1}{|\mathcal{K}_i|} \sum_{k \in \mathcal{K}_i} \mathbf{W}_{\cdot k}. \qquad (5)$$

Thus, this approach leverages the average of the label embeddings associated with $\mathbf{X}_i$, providing an improved representation that captures the intrinsic correlations within the label set.

*4.1.2   Sentence-attentive Prototypical Label Embedding.* The aforementioned average prototypical label embedding approach equally weights each label for a given sentence, while different labels may contribute differently to the classification of that sentence. This inspires us to assign different weights to individual label representations based on their relationship with the sentence when constructing prototypical label embedding.

To achieve the above objective, we analyze the MLTC learning process, where establishing the semantic connection between labels and sentences is crucial. Previous models [4, 6, 29, 33, 35] use the attention mechanism to construct label-specific text representations.

**Figure 2: The overall process of ProtoMix. The orange and green colors of inputs and labels indicate two samples. Gradient colors represent the Mixup operation of input feature space or prototypical label embedding space.**

In particular, they integrate the label information into text representations, ensuring that the relative semantic relationships are preserved. These representations are then utilized for classification tasks. The effectiveness and adaptability of this approach have been validated in many relevant studies, suggesting its potential to also enhance the generation of better prototypical label embeddings.

Building on this, we propose a sentence-attentive method to enhance the prototypical label embedding by strengthening the semantic relationship between the sentence and its corresponding ground-truth label. Specifically, to extract the corresponding semantics related to each sentence, we utilize a sentence-guided attention mechanism to learn the prototypical label embedding. After calculating the cosine similarity between each sentence embedding with its associated multiple label embeddings, we utilize the softmax function to normalize these similarities to obtain the relative semantic relationship weights, and then generate the attentive prototypical label embedding by combining these weights with the label embeddings. The overall process is formalized as follows:

$$\mathbf{K}_i^{[k]} = \frac{\exp g(\mathbf{S}_i, \mathbf{W}_{\cdot k})}{\sum_{k' \in \mathcal{K}_i} \exp g(\mathbf{S}_i, \mathbf{W}_{\cdot k'})}, \tag{6}$$

$$\mathbf{P}_i = \sum_{k \in \mathcal{K}_i} \mathbf{K}_i^{[k]} \mathbf{W}_{\cdot k}, \tag{7}$$

where $g(\cdot, \cdot)$ denotes the cosine similarity between the sentence embedding $\mathbf{S}_i$ and the label embedding $\mathbf{W}_{\cdot k'}$, and $\mathbf{P}_i$ is the weighted sum of the embeddings of all related labels.

## 4.2 Prototypical Label Regularization

Previously, we project the multiple labels into a single point, i.e. the prototypical label embedding, within a dense and compact latent label semantic space. Inspired by the idea that a label represents a view of a sentence, and that multiple labels in MLTC represent multiple views of the sentence [3], we assume that the prototypical label embedding serves as the centroid of these views for a specific sentence. Hence, the sentence embedding should be aligned with its corresponding prototypical label embedding in the latent space.

To achieve this alignment, we apply $L_2$ regularization (i.e. MSE loss) on the distance between the prototypical label embedding $\mathbf{P}_i$ and the sentence embedding $\mathbf{S}_i$. We refer to this constraint as prototypical label regularization, which is defined as follows:

$$\mathcal{L}_M(\mathbf{S}_i, \mathbf{P}_i) = \|\mathbf{S}_i - \mathbf{P}_i\|_2^2 = \frac{1}{H} \sum_{h=1}^{H} (\mathbf{S}_i^{[h]} - \mathbf{P}_i^{[h]})^2. \tag{8}$$

To further explore the impact of MSE loss on prototypical label embeddings and to model the relationships between co-occurrence labels, we conveniently take average prototypical label embedding into account. Through substituting the average prototypical embedding from Eq. 5 into Eq. 8, and normalizing the sentence embedding $\mathbf{S}_i$ and label embedding $\mathbf{W}_{\cdot k}$, the MSE loss expands to:

$$\mathcal{L}_M(\mathbf{S}_i, \mathbf{P}_i) = \left\| \mathbf{S}_i - \frac{1}{|\mathcal{K}_i|} \sum_{k \in \mathcal{K}_i} \mathbf{W}_{\cdot k} \right\|_2^2$$
$$= 1 - \frac{2}{|\mathcal{K}_i|} \mathbf{S}_i^\top \sum_{k \in \mathcal{K}_i} \mathbf{W}_{\cdot k} + \frac{1}{|\mathcal{K}_i|^2} \left\| \sum_{k \in \mathcal{K}_i} \mathbf{W}_{\cdot k} \right\|_2^2. \tag{9}$$

From this equation, there are only two terms left for $\mathcal{L}_M$, since the first term $\|\mathbf{S}_i\|^2 = 1$ as $\mathbf{S}_i$ is normalized. For the second term $\mathbf{S}_i^\top \sum_{k \in \mathcal{K}_i} \mathbf{W}_{\cdot k}$, it approximately has the same optimization direction as part of BCE loss, which force the co-occurring labels $\mathbf{W}_{\cdot k}$ all align to sentence embedding $\mathbf{S}_i$. In contrast, minimizing $\|\sum_{k \in \mathcal{K}_i} \mathbf{W}_{\cdot k}\|_2^2$ leads to the labels repelling each other, therefore, encouraging the diversity of co-occurring labels and preventing over-alignment problem.

More precisely, notice that the second term in Eq.9, similar to the BCE loss, first drives the ground-truth labels to align with the sentence embedding and clusters the embeddings of co-occurring labels together. Then it pushes incorrect labels away from this cluster. However, there is no explicit mechanism to prevent the co-occurring labels from converging too closely, which could lead to an over-alignment issue. Consequently, the discriminative relationships among co-occurring labels should be implicitly learned through iterative training with sufficient training instances.

The last term of MSE loss in Eq.9 emphasizes the diversity of label embeddings within the same cluster by explicitly separating these labels. By imposing MSE loss on both the prototypical label embedding and the sentence embedding, we maintain explicit label correlations and increase the spread of co-occurring labels. Compared to using only BCE loss, incorporating MSE loss regularization introduces a trade-off between alignment and diversity in the representation of co-occurring labels, thereby facilitating the preservation of explicit label correlations.

The prototypical label regularization can be added to $\mathcal{L}_C$ in Eq.3 using a hyper-parameter $\beta$:

$$\mathcal{L}_P = \mathcal{L}_C + \beta \mathcal{L}_M. \tag{10}$$

Later on in our experiments, we refer to this prototypical label regularization model as 'Proto' (or +Proto), which uses $\mathcal{L}_P$ as its loss function for optimization.

### 4.3 Prototypical Label Mixup

*4.3.1 Vanilla Mixup for MLTC Task.* Mixup [34] is a data augmentation and data-dependent regularization approach, which applies linear interpolation in both the input feature space and the corresponding label space. It can be formulated as:

$$\widetilde{\mathbf{X}} = (1-\lambda)\mathbf{X}_i + \lambda\mathbf{X}_j,$$
$$\widetilde{\mathbf{Y}} = (1-\lambda)\mathbf{Y}_i + \lambda\mathbf{Y}_j, \tag{11}$$

where the interpolation ratio $\lambda$ is drawn from a Beta distribution with parameter $\alpha$, denoted as $\lambda \sim \text{Beta}(\alpha, \alpha)$.

When applied to MLTC, vanilla Mixup generates new training instances by linearly interpolating the features $\mathbf{X}_i$ and $\mathbf{X}_j$ and their corresponding multi-hot labels $\mathbf{Y}_i$ and $\mathbf{Y}_j$ using the same interpolation ratio. In our experiments, we refer to this technique as 'Mix' (or +Mix) and denote $\mathcal{L}_{C\_Mix}$ as its loss function:

$$\mathcal{L}_{C\_Mix} = \mathcal{L}_C(\widetilde{\mathbf{Y}}, \hat{\mathbf{Y}}(\widetilde{\mathbf{X}})). \tag{12}$$

*4.3.2 Prototypical Label Embedding Mixup.* Given prototypical embeddings $\mathbf{P}_i$ and $\mathbf{P}_j$ of two sentences $\mathbf{X}_i$ and $\mathbf{X}_j$, with a mixing ratio $\lambda$, Mixup on prototypical embeddings can be naturally defined as:

$$\widetilde{\mathbf{S}} = (1-\lambda)\mathbf{S}_i + \lambda\mathbf{S}_j,$$
$$\widetilde{\mathbf{P}} = (1-\lambda)\mathbf{P}_i + \lambda\mathbf{P}_j, \tag{13}$$

where $\widetilde{\mathbf{S}}$ is the mixed sentence embedding, and $\widetilde{\mathbf{P}}$ is the mixed prototypical label embedding. By generating in-between samples, this approach captures the implicit semantic correlations between labels associated with different sentences. Then, the loss for the mixed sentence embedding and its corresponding mixed prototypical label embedding is defined as:

$$\mathcal{L}_{M\_Mix} = \mathcal{L}_M(\widetilde{\mathbf{S}}, \widetilde{\mathbf{P}}). \tag{14}$$

*4.3.3 ProtoMix for MLTC Task.* Our general Mixup method, ProtoMix, mixes both multi-hot label and distributed prototypical label representation to preserve label correlations. Therefore, the overall loss function contains two components, formulated as:

$$\mathcal{L} = \mathcal{L}_{C\_Mix} + \beta \mathcal{L}_{M\_Mix}. \tag{15}$$

In our experiments, we refer to this approach as ProtoMix, which uses $\mathcal{L}$ as its loss function.

## 5 Experiment

### 5.1 Datasets and Evaluation Metrics

We evaluate our model on the three most widely-used benchmarking MLTC datasets, including **EUR-Lex** [20], **AAPD** [31] and **RCV1** [16]. The statistics of these datasets are shown in Table 1.

We follow previous works [19, 21, 29, 30, 33] on MLTC and adopt six widely used metrics, including P@K and nDCG@K (K = 1,3,5), to evaluate the model performance. It is worth noting that P@1 and nDCG@1 are equal, so we omit nDCG@1 in experiments.

**Table 1: Statistics of three benchmark datasets. $N$ is the number instance, $L$ is the number of label, $\overline{L}$ is the average number of label per instance, and $\overline{I}$ is the average number of instances per label.**

| | $N_{train}$ | $N_{test}$ | $L_{train}$ | $L_{test}$ | $\overline{L}_{train}$ | $\overline{L}_{test}$ | $\overline{I}_{train}$ |
|---|---|---|---|---|---|---|---|
| RCV1 | 23149 | 781265 | 101 | 103 | 3.18 | 3.24 | 729.67 |
| AAPD | 54840 | 1000 | 54 | 54 | 2.41 | 2.42 | 2444.31 |
| EUR-Lex | 15449 | 3865 | 3801 | 2628 | 5.32 | 5.29 | 21.64 |

### 5.2 Baseline Models

To demonstrate the effectiveness of our proposed model, we adopt several representative baseline models for comparison, including **XML-CNN** [19], **SGM** [31], **DXML** [36], **Rank-AE** [4], **AttentionXML** [33], **EXAM** [6], **LSAN** [29], **EHTTN** [30], **LDGN** [21], and **VCLDL** [37]. Detailed introductions of these baselines are shown in Appendix A.

### 5.3 Implementation Details

For the main comparison, we use the pre-trained BERT-base-uncased[1] model with the default initialization network parameters as the encoder. AdamW [13] is exploited as the optimizer, with its learning rates chosen from [0.0001, 0.00002]. The Mixup rate $\alpha$ is chosen from [0.05, 1.5] and the ProtoMix's hyper-parameter $\beta$ is chosen from [0.0005, 0.05]. Note that the small value of $\beta$ does not indicate that $\mathcal{L}_{M\_Mix}$ is not important, but due to the numerical scales of the MSE loss is much larger than the BCE loss.

To comprehensively investigate the adaptability of ProtoMix, we also implement models based on other two representative encoders, CNN and BiLSTM. For both CNN and BiLSTM models, we use the 300-dimension pre-trained GloVe embedding [24] to initialize the word embedding, and choose Adam as the optimizer with learning rates from [0.0001, 0.0005] for different datasets. CNN-based models contain 256 filters with filter sizes ranging from [2, 4, 8]. BiLSTM-based models set hidden dimensions as 512 and the number of layers as 2.

Besides, we observe that supervised fine-tuning large language models have demonstrated convincing performance on multi-class text classification [18]. Following that, we fine-tune the typical large language model LLaMA-3-8B [7] by using the last hidden state as the sentence embedding to train the model. And we utilize LoRA [11], with the rank selected from {16, 32, 64}, for parameter-efficient fine-tuning.

---

[1]https://github.com/huggingface/transformers

**Table 2: Overall performance of ProtoMix. Since the comparison metrics are different from the original paper, for XML-CNN, DXML, SGM and EXAM, the results are borrowed from [29]. The results in italics represent that we reproduced by published code from [18, 33]. And other compared results are borrowed from its original paper. The bolded results are the best performance for each metric.**

| EUR-Lex | | | | | |
|---|---|---|---|---|---|
| Model | P@1 | P@3 | P@5 | nDCG@3 | nDCG@5 |
| XML-CNN | 70.40 | 54.98 | 44.86 | 58.62 | 53.10 |
| SGM | 70.45 | 60.37 | 43.88 | 60.72 | 55.24 |
| DXML | 75.53 | 60.13 | 48.65 | 63.96 | 57.60 |
| Rank-AE | 79.52 | 65.14 | 53.18 | 68.76 | 62.33 |
| *AttentionXML* | *85.87* | *73.85* | *61.76* | *77.08* | *71.22* |
| EXAM | 74.40 | 61.93 | 50.98 | 65.12 | 59.43 |
| LSAN | 79.17 | 64.99 | 53.67 | 68.32 | 62.47 |
| EHTTN | 81.14 | 67.62 | 56.38 | 70.89 | 64.42 |
| LDGN | 81.03 | 67.79 | 56.36 | 71.81 | 66.09 |
| VCLDL | 85.38 | 72.13 | 60.06 | 75.53 | 69.67 |
| ProtoMix | **87.75** | **74.86** | **62.15** | **78.34** | **72.03** |

| AAPD | | | | | |
|---|---|---|---|---|---|
| Model | P@1 | P@3 | P@5 | nDCG@3 | nDCG@5 |
| XML-CNN | 74.38 | 53.84 | 37.79 | 71.12 | 75.93 |
| SGM | 75.67 | 56.75 | 35.65 | 72.36 | 75.35 |
| DXML | 80.54 | 56.30 | 39.16 | 77.23 | 80.99 |
| Rank-AE | - | - | - | - | - |
| *AttentionXML* | *84.90* | *61.06* | *41.76* | *80.42* | *84.43* |
| EXAM | 83.26 | 59.77 | 40.66 | 79.10 | 82.79 |
| LSAN | 85.28 | 61.12 | 41.84 | 80.84 | 84.78 |
| EHTTN | 83.84 | 59.92 | 40.79 | 79.27 | 82.67 |
| LDGN | 86.24 | 61.95 | 42.29 | **83.32** | **86.85** |
| VCLDL | 86.40 | 62.33 | 42.16 | 82.15 | 85.53 |
| ProtoMix | **86.83** | **62.72** | **42.75** | 82.67 | 86.49 |

| RCV1 | | | | | |
|---|---|---|---|---|---|
| Model | P@1 | P@3 | P@5 | nDCG@3 | nDCG@5 |
| XML-CNN | 95.75 | 78.63 | 54.94 | 89.89 | 90.77 |
| SGM | 95.37 | 81.36 | 53.06 | 91.76 | 90.69 |
| DXML | 94.04 | 78.65 | 54.38 | 89.83 | 90.21 |
| Rank-AE | 90.90 | 72.82 | 52.05 | 89.29 | 89.75 |
| *AttentionXML* | *96.75* | *82.36* | *57.40* | *93.18* | *93.90* |
| EXAM | 93.67 | 75.80 | 52.73 | 86.85 | 87.71 |
| LSAN | 96.81 | 81.89 | 56.92 | 92.83 | 93.43 |
| EHTTN | 95.86 | 78.92 | 55.27 | 89.61 | 90.86 |
| LDGN | 97.12 | 82.26 | 57.29 | 93.80 | **95.03** |
| VCLDL | - | - | - | - | - |
| ProtoMix | **97.48** | **83.24** | **57.82** | **94.12** | 94.64 |

We ran experiments five times using different random seeds and reported the average results for comparison. All evaluating experiments are completed on NVIDIA V100, costing about 500s per epoch for BERT for the medium-scale training dataset RCV1.

## 5.4 Overall Performance

The overall performance on three benchmark datasets is reported in Table 2. As the absence of datasets AAPD and RCV1 in the original AttentionXML paper, we re-run and report the results of AttentionXML. For a fair comparison, we don't adopt the model ensemble strategy for AttentionXML while re-producing.

It can be observed that ProtoMix outperforms baseline models on 12 metrics out of 15 metrics. Despite ProtoMix achieving lower performance than the SOTA LDGN model on nDCG@3 and nDCG@5 of AAPD and nDCG@5 of RCV1, ProtoMix achieves larger improvements on other metrics, especially on the EUR-Lex dataset. These results confirm the efficiency of our mixing framework on prototypical label embedding for the MLTC task.

## 5.5 Analysis of Parameters

*5.5.1 Sensitivity of $\alpha$.* $\alpha$ is the important parameter of Mixup that controls Beta Distribution, which further influences the Mixup ratio $\lambda$. To evaluate the impact of $\alpha$, we pick $\alpha$ from a wide range {0.05, 0.1, 0.5, 1, 1.5} and list the performance on EUR-Lex in Table 3. From the Table, we find that under most parameter settings, ProtoMix shows improvements relative to the Base model, which further demonstrates the stability of ProtoMix.

*5.5.2 Sensitivity of $\beta$.* To observe the impact of $\beta$ for the overall loss $\mathcal{L}$, we choose $\beta$ from {0.0001, 0.0005, 0.001, 0.005, 0.01} for comparison. The results of EUR-Lex are reported in Figure 3. As shown in the Figure, the overall performance of Mix surpasses the Base model, which confirms the effectiveness of Mix. The performances of Proto and ProtoMix first increase and then decrease when varying $\beta$. The sweet points are acquired at 0.001 or 0.005, and all of them surpass their respective baselines. This phenomenon shows that $\beta$ affects the results and the MSE-based loss $\mathcal{L}_{M\_Mix}$ contributes to the overall loss $\mathcal{L}$.

**Table 3: Results of ProtoMix on EUR-Lex via different $\alpha$.**

| $\alpha$ | P@1 | P@3 | P@5 | nDCG@3 | nDCG@5 |
|---|---|---|---|---|---|
| 0.05 | 86.36 | 74.29 | 61.76 | 77.52 | 71.35 |
| 0.1 | 86.62 | 74.32 | 62.12 | 77.64 | 71.71 |
| 0.5 | 87.37 | 74.99 | 62.39 | 78.35 | 72.17 |
| 1 | 87.81 | 75.05 | 62.14 | 78.46 | 72.04 |
| 1.5 | 87.94 | 74.70 | 62.16 | 78.21 | 72.00 |

## 5.6 Analysis of Ablation Study

Since ProtoMix is a framework that can be directly employed in many basic MLTC models with different encoders, to evaluate the flexibility and adaptability of the framework, we choose two other representative encoders, CNN and BiLSTM, and a popular decoder, LLaMA, as the base model (described in Section 5.3). Moreover, to evaluate the effectiveness of different variants, we conduct ablation studies for ProtoMix based on these basic models. Specially, the comparison includes Base (the basic model for each encoder, or decoder), +Proto (Base model enhanced by the prototypical label embedding regularization method), +Mix (Base model enhanced by vanilla Mixup method), and +ProtoMix (Base model enhanced

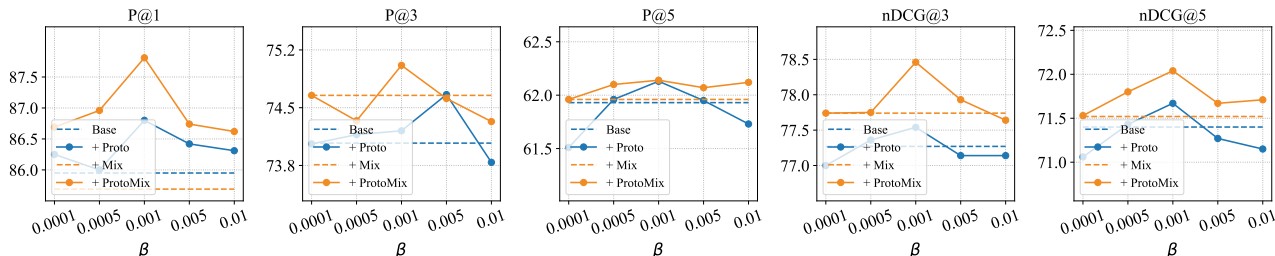

Figure 3: Performance on EUR-Lex via different $\beta$.

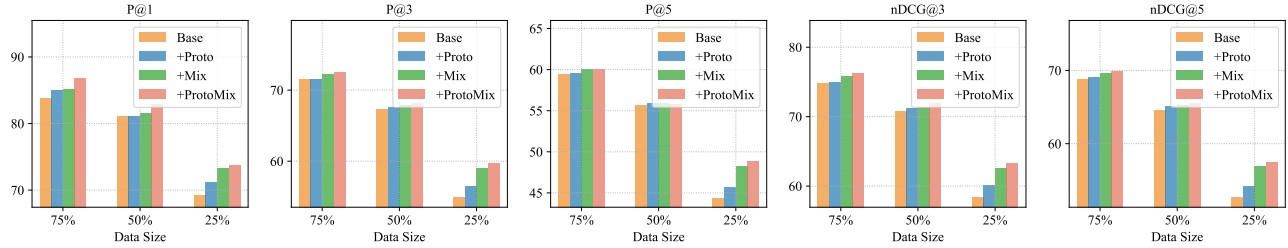

Figure 4: Performance on sparse data of EUR-Lex via different downsampling ratios.

Table 4: Performance on diverse basic models and corresponding ablation study of EUR-Lex.

| Method | P@1 | P@3 | P@5 | nDCG@3 | nDCG@5 |
|---|---|---|---|---|---|
| CNN | 74.75 | 60.89 | 50.23 | 64.39 | 58.91 |
| +Proto | 76.35 | 62.23 | 51.77 | 65.81 | 60.53 |
| +Mix | 75.55 | 63.05 | 52.14 | 66.35 | 60.86 |
| +ProtoMix | **77.10** | **63.42** | **52.96** | **66.88** | **61.68** |
| BiLSTM | 78.78 | 65.30 | 54.13 | 68.77 | 63.06 |
| +Proto | 81.91 | 68.38 | 56.26 | 71.93 | 65.71 |
| +Mix | 79.51 | 66.91 | 55.99 | 70.13 | 64.77 |
| +ProtoMix | **82.87** | **69.31** | **57.30** | **72.78** | **66.69** |
| BERT | 85.95 | 74.04 | 61.93 | 77.27 | 71.40 |
| +Proto | 86.59 | 74.54 | 61.97 | 77.82 | 71.63 |
| +Mix | 85.69 | 74.66 | 61.96 | 77.74 | 71.52 |
| +ProtoMix | **87.75** | **74.86** | **62.15** | **78.34** | **72.03** |
| LLaMA | 77.32 | 63.19 | 51.47 | 66.85 | 60.53 |
| +Proto | **79.35** | 63.55 | 51.96 | 67.49 | 61.20 |
| +Mix | 78.12 | 64.01 | 52.10 | 67.68 | 61.27 |
| +ProtoMix | 79.09 | **64.25** | **52.26** | **68.04** | **61.55** |

Table 5: Comparison of different data augmentation methods.

| Method | P@1 | P@3 | P@5 | nDCG@3 | nDCG@5 |
|---|---|---|---|---|---|
| Base | 85.95 | 74.04 | 61.93 | 77.27 | 71.40 |
| +EDA | 85.12 | 73.89 | 61.82 | 76.99 | 71.13 |
| +BT | 85.25 | 72.94 | 60.87 | 76.31 | 70.40 |
| +LLM | 86.00 | 73.85 | 61.49 | 77.13 | 71.03 |
| +ProtoMix | **87.75** | **74.86** | **62.15** | **78.34** | **72.03** |

by prototypical label embedding Mixup method). To differentiate, we employ specific names of basic models, such as CNN, BiLSTM, BERT, LLaMA, instead of the generic term Base. The results on EUR-Lex are shown in Table 4, and results on AAPD and RCV1 are shown in Appendix B.

From the Table, we find that all variants surpass the Base model in the majority of comparable metrics, which proves the effectiveness of these regularization methods. Comparing all Proto models with their corresponding Base models, or ProtoMix models with their corresponding Mix models, Proto and ProtoMix achieve better performance on almost all evaluation metrics. This indicates that our presented MSE-based prototypical label regularization is effective. Additionally, ProtoMix models consistently outperform Proto and Mix on all datasets and various basic models. These results demonstrate the superiority and indicate the adaptability of our prototypical label embedding Mixup framework.

It is worth noting that LLaMA performs somewhat worse than BERT on EUR-Lex, but delivers comparable results on AAPD and RCV1. More precisely, as the number of labels increases in a dataset, LLaMA's performance worsens. We guess that this is because MLTC tasks rely more heavily on the expressive power of sentence embeddings than multi-class text classification does, posing challenges for LLMs with a decoder architecture. This issue becomes increasingly pronounced with a higher number of labels, leading to notably poor performance on EUR-Lex, which contains the most labels.

## 5.7 Analysis of Model Regularization

*5.7.1 Performance on Sparse Data.* As ProtoMix is a regularization method, to further analyze the robustness in sparse settings, we downsample the training data with ratio 75%, 50%, 25% for evaluation. Specifically, we use the sparse datasets training these regularization methods and report the results on EUR-Lex in Figure

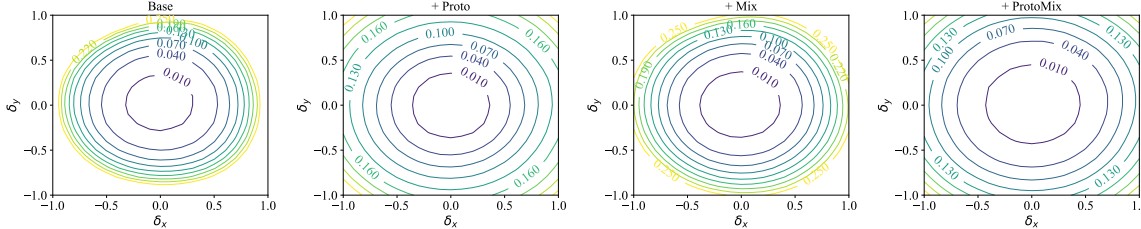

Figure 5: The 2D visualization of the minima of the empirical risk selected by four variants on EUR-Lex.

4, AAPD and RCV1 in Appendix C. From the Figure, we can observe that almost all regularization methods show improvements in the sparse dataset. As the dataset size decreases, all performance tends to decline. However, the decrease for ProtoMix is smaller compared to Proto and Mix. This observation shows the effectiveness of our prototypical label Mixup framework.

*5.7.2 Comparison of Data Augmentation Methods.* Since Mixup is also a data augmentation method, we evaluate three popular data augmentation techniques, easy data augmentation (EDA), back translation (BT), and large language model (LLM) for comparison. For LLM, we use 'Rewrite the sentence: ' as the prompt based on ChatGPT `gpt-3.5-turbo` [22]. We implement these methods based on the same Base model consistent with ProtoMix. We report the results on EUR-Lex in Table 5, and AAPD and RCV1 in Appendix C. From Table 5, we find ProtoMix achieves better performance than other plug-and-play methods, which demonstrates the effectiveness of our proposed data augmentation method.

*5.7.3 Visualization of Loss Landscape.* In order to evaluate the generalization ability of the models, we adopt visualization technique [17] to analyze the loss landscape around the minima of the empirical risk selected by the comparable methods. Specially, we compute the landscape by:

$$\mathcal{L}(\theta^* + \delta_x \mathbf{d}_x + \delta_y \mathbf{d}_y). \tag{16}$$

where $\mathbf{d}_x$ and $\mathbf{d}_y$ are the random directions of the optimal parameter $\theta^*$, and $\delta_x$ and $\delta_y$ are the step sizes along with $\mathbf{d}_x$ and $\mathbf{d}_y$ respectively. It has been shown in [17] that flatter minima and wider regions of apparent convexity imply better generalization.

The 2D visualization of the training set of EUR-Lex are shown in Figure 5, and AAPD and RCV1 are shown in Appendix C. The sparser the contours in the central area are, the flatter the minima are. We observe that Proto, Mix, and ProtoMix achieve better generalization than the Base model. This result demonstrates that all compared regularization approaches improve MLTC performance. We then discover that Proto and ProtoMix have flatter minima than Base and Mix respectively. This confirms the effectiveness of our proposed prototypical label regularization. We finally find ProtoMix has the flattest minima compared with other variants shown in Figure 5, which indicates ProtoMix achieves the best generalization capability among all variants.

## 5.8 Analysis of Label Correlation

Singular Value Decomposition (SVD) [1] is basically a matrix factorization technique, which is widely used in data dimensionality

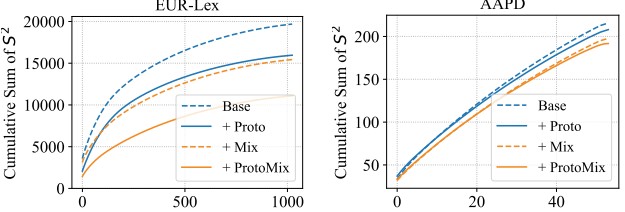

Figure 6: The diversity of label embedding matrix. $S$ represents the singular value.

reduction. The magnitude of the singular values reveals the statistical significance, or, the correlation between data. Therefore, we adopt SVD on the label embedding matrix to study the diversity of label embedding for verifying the tolerance of over-alignment by these regularization methods. We rank the singular values from high to low and plot the cumulative sum of squares of the values in Figure 6. The slower the curve increases, the more diverse the label embedding vectors that our model learns.

As shown in Figure 6, Proto/ProtoMix performs better than Base/Mix on all datasets respectively, indicating that the prototypical label embedding regularization mitigates the over-alignment issue and behaves better in prompting label diversity. We also find that the gap between Proto/ProtoMix and Mix/Base of EUR-Lex is larger than AAPD. The reason might be that EUR-Lex has more labels and fewer instances for each label compared with AAPD, leading to over-alignment being more severe in EUR-Lex. Furthermore, ProtoMix exhibits the slowest rate of increase, demonstrating that the label embeddings it learns are more diverse. This indicates that ProtoMix is more effective at mitigating the over-alignment issue and, consequently, better at preserving label correlation.

## 6 Conclusion and Future Work

To alleviate the overfitting problem and preserve explicit and implicit label correlation on MLTC, we first generate a sentence-attentive prototypical label embedding as a bridge. We then present a prototypical label regularization between prototypical label embedding and sentence embedding to preserve explicit co-occurring label correlation. We finally propose a prototypical label embedding Mixup method to preserve implicit semantic label correlation. Empirical studies confirm the effectiveness of our model. In future work, we may adapt our regularization method to improve the generalization of the extreme multi-label text classification, where labels are usually sparse and long-tailed distributed.

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

# A  Baselines

**XML-CNN** [19] adopts multiple convolutional kernels and dynamic pooling to extract high-level sentence features for MLTC. This approach also considers multi-label co-occurrence patterns in both the optimization objective and the network architecture.

**SGM** [31] formulates MLTC as a sentence generation model and takes relation correlation and word contribution into account. It simultaneously captures the correlations between labels and selects the most informative words while predicting different labels.

**DXML** [36] introduces explicit label co-occurrence graph to MLTC and uses deep walk algorithm to encode label embedding on this graph for label interaction. It also establishes non-linear embedding in both feature and label spaces to increase the performance.

**Rank-AE** [4] combines word-vector-based self-attention with Auto-Encoder to model the inter-label dependencies and the feature-label dependencies simultaneously. And it exploits a margin-based ranking loss for efficient training and noisy labeled data handling.

**AttentionXML** [33] utilizes the attention mechanism to capture the label-specific information and generate a shallow and wide probabilistic label tree to handle millions of labels. We focus on standard MLTC rather than extreme MLTC, thus eliminating label clustering and model ensemble in comparison.

**EXAM** [6] treats the parameters of the classifier as label embedding and utilizes attention to characterize the relation between words and classes. It leverages the interaction mechanism to explicitly compute the word-level interaction signals for text classification.

**Table 6: Performance on diverse basic models and corresponding ablation study.**

| | AAPD | | | | |
|---|---|---|---|---|---|
| Method | P@1 | P@3 | P@5 | nDCG@3 | nDCG@5 |
| CNN | 82.90 | 59.60 | 41.10 | 78.77 | 83.04 |
| +Proto | 83.90 | 60.23 | 40.56 | 79.61 | 82.86 |
| +Mix | 83.50 | 60.70 | 40.82 | 79.82 | 83.04 |
| +ProtoMix | **84.60** | **61.63** | **41.26** | **81.00** | **84.09** |
| BiLSTM | 83.00 | 60.03 | 41.26 | 79.11 | 83.23 |
| +Proto | 83.90 | 60.13 | 41.10 | 79.50 | 83.44 |
| +Mix | 84.89 | 60.69 | **41.82** | 80.21 | 84.47 |
| +ProtoMix | **85.40** | **61.40** | 41.66 | **81.17** | **84.80** |
| BERT | 84.90 | 61.60 | 42.11 | 80.99 | 84.99 |
| +Proto | 85.20 | 61.90 | 42.53 | 81.57 | 85.85 |
| +Mix | 85.50 | **62.76** | 42.54 | 82.49 | 86.15 |
| +ProtoMix | **86.83** | 62.72 | **42.75** | **82.67** | **86.49** |
| LLaMA | 87.90 | 64.69 | **43.72** | 84.91 | 88.46 |
| +Proto | 88.70 | 64.43 | 43.46 | 84.81 | 88.35 |
| +Mix | 88.50 | 64.86 | 43.44 | 85.19 | 88.43 |
| +ProtoMix | **89.10** | **65.06** | 43.52 | **85.45** | **88.59** |
| | RCV1 | | | | |
| CNN | 94.93 | 78.21 | 54.91 | 89.15 | 90.34 |
| + Proto | 95.21 | 78.52 | 54.93 | 89.51 | 90.53 |
| + Mix | 94.84 | 77.84 | 54.57 | 88.85 | 89.90 |
| + ProtoMix | **95.54** | **79.28** | **55.23** | **90.29** | **91.07** |
| BiLSTM | 94.88 | 78.36 | 54.90 | 89.29 | 90.34 |
| + Proto | 95.33 | 78.42 | 54.87 | 89.51 | 90.50 |
| + Mix | 94.49 | 78.31 | 54.84 | 89.17 | 90.20 |
| + ProtoMix | **95.63** | **79.35** | **55.32** | **90.35** | **91.16** |
| BERT | 97.02 | 83.22 | 57.87 | 93.94 | 94.53 |
| +Proto | 97.41 | 83.19 | 57.81 | 94.01 | 94.55 |
| +Mix | 96.67 | **83.26** | **57.95** | 93.97 | 94.58 |
| +ProtoMix | **97.48** | 83.24 | 57.82 | **94.12** | **94.64** |
| LLaMA | 96.33 | 80.71 | 56.44 | 91.65 | 92.62 |
| +Proto | **96.49** | 80.97 | 56.61 | 92.01 | 92.91 |
| +Mix | 96.41 | 81.02 | 56.65 | 92.07 | 92.96 |
| +ProtoMix | 96.31 | **81.17** | **56.74** | **92.13** | **93.01** |

**LSAN** [29] leverages label semantic information to establish the relationship between labels and documents, and further incorporates a self-attention mechanism to identify label-specific document representations based on the content information of the documents.
**EHTTN** [30] transfers the meta-knowledge of high-frequency labels to low-frequency labels for improving the long-tail label representation performance.
**LDGN** [21] integrates category information to extract label-specific components from documents. It utilizes a dual Graph Convolution Network (GCN) to capture comprehensive and adaptive interactions among these components.

**Table 7: Comparison of different data augmentation methods on AAPD and RCV1.**

| | AAPD | | | | |
|---|---|---|---|---|---|
| Method | P@1 | P@3 | P@5 | nDCG@3 | nDCG@5 |
| Base | 84.90 | 61.60 | 42.11 | 80.99 | 84.99 |
| +EDA | 86.70 | 62.06 | 42.46 | 82.23 | 86.26 |
| +BT | 85.60 | 61.90 | 42.38 | 81.61 | 85.72 |
| +LLM | 85.40 | 62.20 | 42.30 | 81.92 | 85.81 |
| +ProtoMix | **86.83** | **62.72** | **42.75** | **82.67** | **86.49** |
| | RCV1 | | | | |
| Base | 97.02 | 83.22 | **57.87** | 93.94 | 94.53 |
| +EDA | 97.12 | 82.44 | 57.34 | 93.35 | 93.97 |
| +BT | 97.27 | 82.60 | 57.40 | 93.51 | 94.07 |
| +LLM | 97.38 | 82.91 | 57.66 | 93.81 | 94.39 |
| +ProtoMix | **97.48** | **83.24** | 57.82 | **94.12** | **94.64** |

**VCLDL** [37] proposes a variational continuous label distribution learning framework, which establishes a theoretical connection between the feature space and the label space, enabling the extraction of hidden information within the observable logical labels.

## B Analysis of Ablation Study on AAPD and RCV1

Performance on diverse encoders and corresponding ablation study for AAPD and RCV1 are shown in Table 6.

## C Analysis of Model Regularization on AAPD and RCV1

Performance on sparsify AAPD and RCV1 are shown in Figure 7. And the comparison of data augmentation methods on AAPD and RCV1 is reported in Table 7. Consistent conclusion with that in Section 5.7 shows the robustness and scalability of ProtoMix.

The 2D visualization on AAPD and RCV1 is shown in Figure 8. ProtoMix achieves the flattest minima compared with Proto and Mix, which is consistent with that in EUR-Lex. Besides, it is worth noting that the generalization performance of EUR-Lex is much better than of AAPD, we think this is because EUR-Lex enjoys larger-scaling labels leading to much diversity of label embedding through regularization methods. This phenomenon can be explained in Section 5.8.

Received 20 February 2007; revised 12 March 2009; accepted 5 June 2009

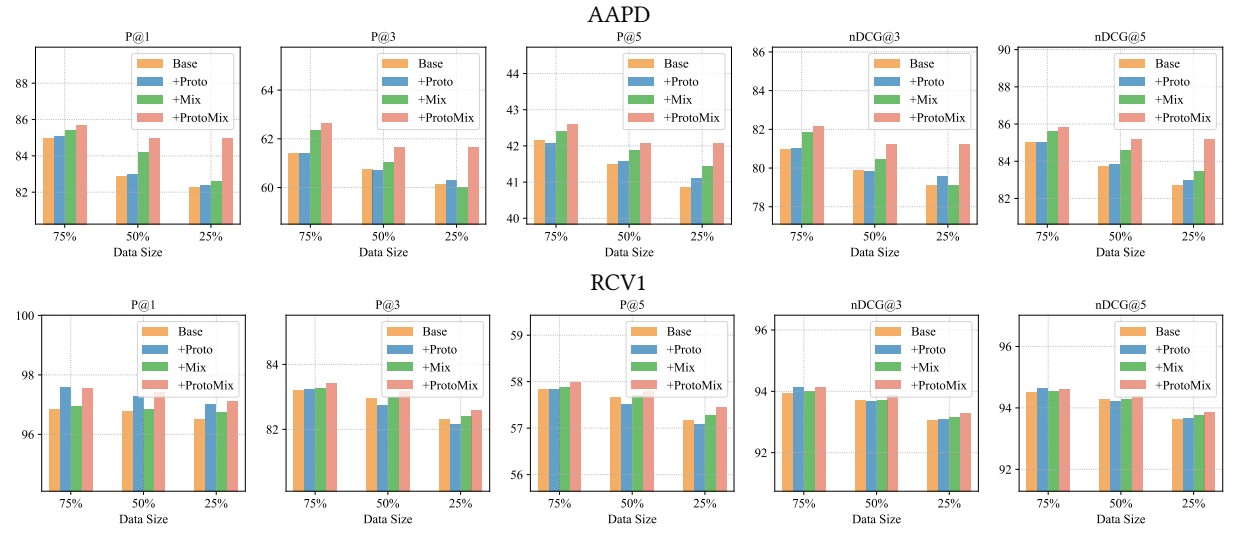

Figure 7: Performance on sparse data via different downsampling ratios.

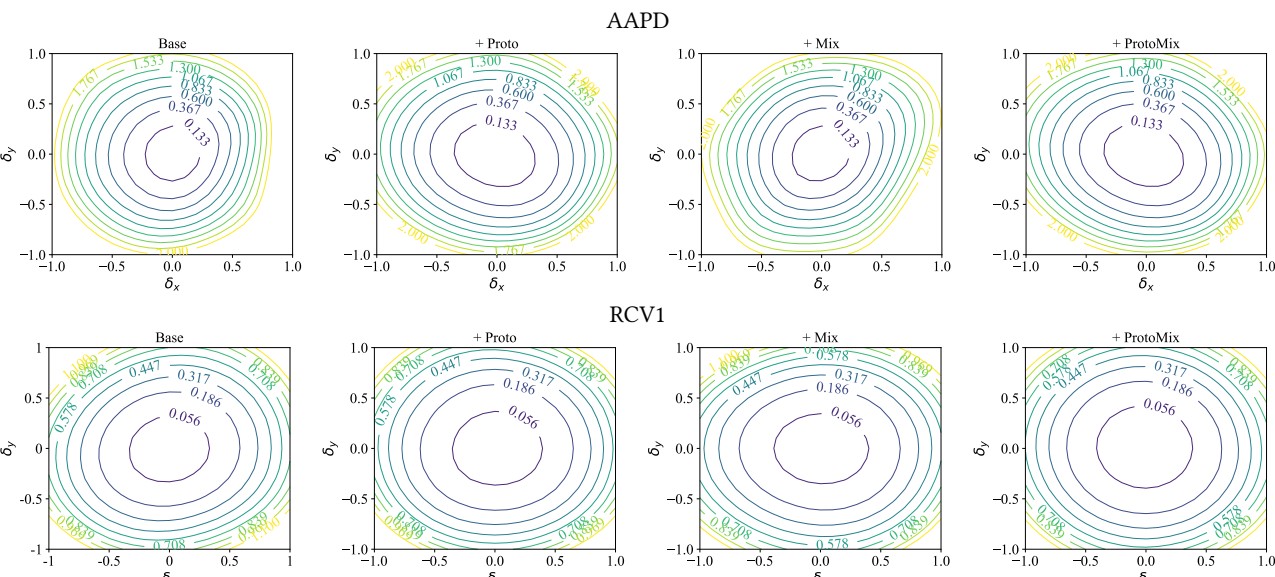

Figure 8: The 2D visualization of the minima of the empirical risk selected by four variants.

