# OpenReview forum: "Preserving Label Correlation for Multi-label Text Classification by Prototypical Regularizations"
_ACM.org/TheWebConf/2025/Conference — WWW 2025 Poster_

### Official Review · Reviewer_624p · 2024-11-27

**Novelty:** 4
**Technical Quality:** 4

**Review:**

The paper **"Preserving Label Correlation for Multi-label Text Classification by Prototypical Regularizations"** introduces methods to improve multi-label text classification (MLTC) by addressing the challenges of preserving explicit (co-occurrence) and implicit (semantic) label correlations while mitigating overfitting. The authors propose:

1. **Prototypical Label Regularization**: Aligns sentence embeddings with prototypical label embeddings (centroids of co-occurring labels) to preserve explicit correlations and prevent over-alignment.

2. **Prototypical Label Mixup**: Extends the Mixup technique by blending prototypical label embeddings to capture implicit correlations across sentences.

These methods form the **ProtoMix** framework, which outperforms state-of-the-art MLTC models on multiple benchmarks (EUR-Lex, AAPD, RCV1) in terms of metrics like precision and nDCG. The framework is robust in sparse data settings and enhances generalization by preserving label diversity.

**Strengths**

* S1: The paper clearly identifies the challenges of **multi-label text classification (MLTC)**, particularly the importance of capturing both **explicit** (co-occurrence-based) and **implicit** (semantic-based) label correlations while mitigating overfitting.

* S2: The paper proposes a novel regularization (i.e. **MSE** loss) on the distance between the prototypical label embedding and the sentence embedding, addressing the over-alignment problem induced by conventional **BCE** loss

* S3: The paper first exploits **Mixup** in MLTC, which applying Mixup on both multi-hot label embedding and prototypical label embedding to preserve the implicit label correlation

* S4: The paper evaluates the proposed methods on standard MLTC benchmarks (e.g., EUR-Lex, AAPD, RCV1) and demonstrates consistent performance improvements across metrics such as Precision@K and nDCG@K.

**Weakness**

* W1: The part **2 Related Work** seems insufficient--there is only simple introduction of other MLTC methods but without any analysis of them (i.e. weakness of these methods) and comparison with the proposed method. Thus, readers can not know the connection and difference between this paper and other works.
* W2: The main contribution of the paper is a novel regularization combining **MSE** loss and **Mixup**, but there is no in-depth analysis of other regularization in MLTC (only a simple introduction of BCE loss in part **3 Preliminaries**)
* W3:  In the part **4.1 Prototypical Label Gerneration**, there is no illustration of difference between proposed attention method and others attention methods. Simply using softmax attention to obtain weights lacks novelty.
* W4: In Section 4.2 Prototypical Label Regularization and Equation 10, why is the combination of BCE loss and $\beta$ MSE loss used? Is the performance of using only MSE loss suboptimal? The experiments lack a direct comparison between BCE loss and MSE loss. Testing BCE and MSE losses separately within the proposed framework would provide better evidence, but the current ablation study only examines different values of $\beta$, which does not adequately demonstrate that MSE loss is superior to BCE loss.
* W5: While EUR-Lex, AAPD, and RCV1 are standard datasets, they may not fully capture real-world MLTC challenges, such as highly imbalanced label distributions or few-shot scenarios, which are more prone to severe overfitting. Evaluating the proposed methods on a broader range of datasets could further strengthen the validity of the claims.

**Questions:**

**Questions**

Q1: The in-depth comparison of proposed regularization and other regularization?

Q2: Why using BCE loss + $\beta$ MSE loss? Is the performance of only MSE loss bad?

Q3:  The difference of proposed attention method and other attention methods?

**Reviewer Confidence:**

3: The reviewer is confident but not certain that the evaluation is correct

**Scope:**

3: The work is somewhat relevant to the Web and to the track, and is of narrow interest to a sub-community

---

### Official Review · Reviewer_Q4Mq · 2024-11-28

**Novelty:** 5
**Technical Quality:** 4

**Review:**

The paper proposes a solution for overfitting and label correlation preservation in MLTC, distinguishing two correlation types and introducing two regularization methods. Evaluation on three datasets with multiple metrics and against various baselines is extensive, with detailed result analyses. But performance on certain metrics of AAPD and RCV1 datasets needs improvement. The paper is generally clear in its presentation, though some theoretical parts could be more accessible. The theoretical contributions are valuable, and ProtoMix shows practical benefits with performance gains over many baselines and useful insights from encoder exploration and sparse data analysis.

Pros:
* Novel approach to handle label correlations in MLTC.
* Comprehensive experimental evaluation with multiple datasets, metrics, and baselines.
* The proposed methods show good performance and effectiveness in many cases.

Cons (detailed in the Questions section):
* Some theoretical parts could be more accessible for a wider readership.

**Questions:**

1. ProtoMix performs well but lags behind LDGN on some AAPD and RCV1 metrics. What causes this gap and how can future research boost its performance on these datasets?
2. You used certain encoders. Have you considered other emerging/alternative ones? If so, what were the findings or reasons for exclusion?
3. Real-world annotation difficulty affects the method. What strategies can mitigate these negative effects?

**Reviewer Confidence:**

2: The reviewer is willing to defend the evaluation, but it is likely that the reviewer did not understand parts of the paper

**Scope:**

4: The work is relevant to the Web and to the track, and is of broad interest to the community

---

### Official Review · Reviewer_2wPJ · 2024-11-29

**Novelty:** 4
**Technical Quality:** 4

**Review:**

**Summary:**

This paper presents ProtoMix, a framework for multi-label text classification (MLTC) that integrates prototypical regularizations to address overfitting and preserve label correlations. ProtoMix uses MSE loss to align sentence embeddings with prototypical label embeddings for explicit correlation preservation and extends Mixup to label embeddings for fostering implicit correlations. Experiments on three benchmark datasets demonstrate its effectiveness, achieving significant improvements over state-of-the-art methods.

**Strengths:**

1.	The authors effectively explain the challenges of over-alignment and correlation preservation in MLTC.
2.	The framework effectively integrates explicit and implicit label correlations, addressing a critical challenge in MLTC by utilizing prototypical regularizations and mixup strategies.
3.	The framework is adaptable across different encoders and maintains performance under sparse data conditions.

**Weaknesses:**

1.	As per the conference requirements, every submission must clearly state on the first page how the work is relevant to the Web and aligned with the chosen track. It is recommended that the authors explicitly describe how this study addresses a Web-related scientific challenge.
2.	The selected baselines in this paper are not very recent, with the most recent ones only from 2023. Could the authors clarify the strategy behind selecting the baselines?
3.	The novelty of the proposed method appears to be limited, as the combination of prototype guidance and mixup techniques to address overfitting has already been explored in prior research, such as in [1], which also introduces a model named ProtoMix.

[1] Xu Y, Jiang X, Chu X, et al. Protomix: Augmenting health status representation learning via prototype-based mixup[C]//Proceedings of the 30th ACM SIGKDD Conference on Knowledge Discovery and Data Mining. 2024: 3633-3644.

**Questions:**

1.	While the approach is effective for the datasets used, its scalability to extreme multi-label problems (e.g., with thousands of labels) is not thoroughly analyzed. Can the framework be scaled effectively to extreme multi-label text classification tasks involving thousands of labels?
2.	Paper [1] also addresses overfitting issues by utilizing prototype-guided representations and mixup strategies. Could the authors elaborate on how does your approach differentiates itself? Are there any fundamental differences in how prototypes are incorporated into the mixup process between your approach and [1]?
3.	Given LLaMA's suboptimal performance compared to BERT on EUR-Lex, did the authors consider fine-tuning larger variants of LLaMA or using other large language models (LLMs)? How might this affect the overall findings?

**Reviewer Confidence:**

3: The reviewer is confident but not certain that the evaluation is correct

**Scope:**

3: The work is somewhat relevant to the Web and to the track, and is of narrow interest to a sub-community

---

### Official Review · Reviewer_zaXM · 2024-12-01

**Novelty:** 6
**Technical Quality:** 6

**Review:**

This paper presents a novel framework, ProtoMix, for multi-label text
classification (MLTC), trying to address both overfitting and label correlation preservation effectively.Extensive experiments on three databases demonstrate ProtoMix's effecitiveness over baseline models.

Strength:
1) The methodological framework is backed by theoretical analysis, explaining the impact of MSE-based loss on reducing over-alignment and promoting diversity among label embeddings.
2) The paper includes a comprehensive analysis, offering valuable insights into the model's behavior under different conditions.
3) The paper is well-structured, with logical of organization of theoretical foundations, methodology and experimental evaluation.
4) The writing style is concise.
5) The integration of prototypical label embeddings with Mixup for preserving explicit and implicit label correlations is an innovative approach that combines ideas from data augmentation and few shot learning.

weaknesses:
1) while the paper demonstrates substantial empirical success, computational efficiency could be a concern due to the additional regularization and embedding mixing processes, which might limit scalability.
2) while the albation study results are detailed, the interpretation of differences between methods needs further clarification.

**Questions:**

1) On the Line 107 of Page 1, the author claims that "...existing training objectives fail to preserve explicit correlation because they use BCE losses", which may be somewhat contradictory to the previous discussion on the Line 67 of Page 1.

2)	The authors try to preserve explicit and implicit label correlations on MLTC by introducing prototype label regularization. However, there is no clear theoretical distinction on whether the regularization is about explicit or implicit correlations or both. Similarly, the effects of explicit and implicit label correlations are not analyzed separately in the experiments.

3) Contrastive learning is a good technique to preserve label correlation and it should be discussed in related work.

4) There is a g function in Figure 2, but it is not introduced in the manuscript.

5) In Figure 6, what does the horizontal axis represent?

**Reviewer Confidence:**

4: The reviewer is certain that the evaluation is correct and very familiar with the relevant literature

**Scope:**

4: The work is relevant to the Web and to the track, and is of broad interest to the community

---

### Official Review · Reviewer_nmfP · 2024-12-03

**Novelty:** 5
**Technical Quality:** 6

**Review:**

The paper introduces a novel framework, ProtoMix, that combines prototypical label regularization and Mixup for preserving explicit and implicit label correlations in multi-label text classification (MLTC). This approach is innovative and effectively addresses the unique challenges of MLTC. The paper identifies and differentiates the two types of label correlations (explicit and implicit) and highlights the shortcomings of existing methods, such as over-alignment issues caused by binary cross-entropy loss.
The experimental evaluation is robust, utilizing three benchmark datasets (EUR-Lex, AAPD, and RCV1). The results convincingly demonstrate the superiority of ProtoMix over several baseline methods across multiple metrics.
Although the paper addresses label diversity, more focused evaluation on long-tailed labels or highly imbalanced datasets would strengthen the claims of robustness.

**Questions:**

1. How the proposed method perform on imbalanced datasets or long-tailed labels?
2. It would be better to have more description of the compared works.

**Reviewer Confidence:**

3: The reviewer is confident but not certain that the evaluation is correct

**Scope:**

3: The work is somewhat relevant to the Web and to the track, and is of narrow interest to a sub-community